# Effects of an integrated intervention on schistosomiasis prevalence in a rural area of Tanzania

**Yoonho Cho[1], Jungim Lee[1], Humphrey Deogratias Mazigo[2], Leah Elisha Salamba[3], Seungman Cha[4]***

**1** World Vision Korea, Yeongdeungpo, Republic of Korea, **2** School of Public Health, Catholic University of Health, and Allied Sciences, Mwanza, Tanzania, **3** World Vision Tanzania, Njiro, Tanzania, **4** Department of Disease Control, London School of Hygiene & Tropical Medicine, London, United Kingdom

* Seungman.Cha@alumni.lshtm.ac.uk

## Abstract

Neglected tropical diseases have a huge impact around the world, particularly in sub-Saharan Africa. The global NTD community and countries have emphasized the importance of integrated interventions. However, findings on effectiveness of integrated interventions have been inconsistent. The study was a quasi-experimental design with repeated cross-sectional data collected from both intervention and control groups in 2020 and 2022. School-aged children from 12 schools were divided into three groups: the school-based mass drug administration (MDA) group, the school-based and community-based MDA group, and the integrated model group. A multilevel logistic regression model was employed to assess the impact of the integrated intervention on schistosomiasis prevalence, with individuals as the first level and groups as the second level. Examining the effect of the intervention on the prevalence of schistosomiasis among students, both the school-based and community-based MDA group (risk difference [RD] -13.7%; 95% confidence interval [CI] -24.5% to -2.9%; p < .05) and the integrated model group (RD -23.3%; 95% CI -32.7% to -13.9%; p < .001) showed significantly greater reductions compared to the school-based MDA alone group. In addition, the integrated model group (RD -9. 6%; 95% CI -23.3% to 4.1%; p > .05) had a greater reduction in infection rates than the school-based and community-based group. For behavior change outcomes, when comparing changes in the proportion of water contact behaviors, the integrated model group had a significantly greater reduction compared to the school-based and community-based group (RD -56.0%; 95% CI -67.3% to -44.8%; p < .05). When comparing the change in home toilet use between groups, the fully integrated model group performed significantly better than the other two groups. This study found that an integrated approach to *S. haematobium* elimination was consistently effective in both students and adults. Further research is required to better understand the effectiveness of integrated NTD control strategies and where improvements are needed.

**Data availability statement:** Data are available in S1 and S2 Data files.

**Funding:** This research was supported by the Korea International Cooperation Agency (KOICA) (Letter of Grant Agreement for KOICA's Public-Private Partnership Projects with Global Disease Eradication Fund No.2020-01; project management to J.L.) and World Vision Korea (No.213612; project management to J.L.) under the title of "Neglected Tropical Diseases Elimination Project in Itilima District, Tanzania" between 2020 and 2022. The funders had no role in study design, data collection and analysis, decision to publish, or preparation of the manuscript.

**Competing interests:** The authors have declared that no competing interests exist.

## Author summary

The effectiveness of interventions that integrate MDAs, WASH and behavior change has been highlighted by the global community to address the NTD problem. Several studies have examined the effectiveness of integrated strategies, but the findings were different. This study aims to determine the effects of integrated interventions on schistosomiasis prevalence in rural Tanzania. In our study, the effectiveness of integrated NTD control models, which included MDA, community-led total sanitation, and health education, was more pronounced than when relying on school-based MDA alone. Furthermore, there was a significant decrease in water contact behavior and an increase in use of improved toilets at home in the integrated model group compared to other groups.

## Introduction

Schistosomiasis is a parasitic disease caused by flatworms of the genus *Schistosoma*, affecting over 251 million people worldwide [1]. The global impact of schistosomiasis remains substantial, with more than 90% of cases found in sub-Saharan Africa [2,3]. Although the disease predominantly affects rural regions, the incidence of schistosomiasis in urban areas is increasing due to urbanization [4–6].

The World Health Organization (WHO) launched its first NTD Roadmap in 2012, aimed at addressing the impact of neglected tropical diseases (NTDs) through a coordinated strategy [7,8]. This initiative has led to significant progress in controlling NTDs [8,9], particularly by contributing to the achievement of 67% coverage of preventive chemotherapy for school-aged children against schistosomiasis globally in 2019 [9,10].

However, despite these efforts, many of the objectives outlined in the initial roadmap for 2020 were not met, leading to the release of a second roadmap in the same year. The updated roadmap for 2021–2030 identified key areas of deficiency and outlined the necessary actions to achieve the 2030 targets. It envisions a future where the burden of NTDs will be significantly reduced, controlled, eliminated, or eradicated by the year 2030 [9].

The main strategies of the WHO roadmap to address schistosomiasis include preventive chemotherapy through mass drug administration (MDA) with praziquantel (PZQ), investment in water and sanitation infrastructure, snail control, health education, and behavior change interventions. The global NTD community has recognized a critical need to strengthen the evidence base regarding the effectiveness of water, sanitation, and hygiene (WASH) and behavior change interventions in promoting treatment compliance and healthy behaviors [9].

The Tanzanian government established the National NTD Control Programme in 2009, aimed at guiding the implementation of the country's NTD control program in alignment with the WHO's approach [11], emphasizing community involvement and collaboration [11–13]. This includes island control programs and nationwide treatment efforts, which have made progress [14,15]. While MDA of PZQ is a primary control measure [16–18], its sustainability is a concern due to reinfection [19].

Accordingly, the Tanzanian government launched the second Tanzanian NTD Master Plan in 2022. The second Tanzanian NTD master plan (2021–2026) targets priority diseases such as lymphatic filariasis (LF), trachoma, onchocerciasis, soil-transmitted helminthiases(STH), and schistosomiasis. It aims to improve access to preventive chemotherapy, integrate control measures, strengthen monitoring systems, and encourage research [12,13]. Additionally, the plan now focuses on sustainable control, emphasizing supplementary strategies beyond MDA, including health education and behavioral interventions [12,13].

Several studies have examined the impact of integrated strategies on NTDs. In their systematic review, Banda and colleagues (2021) analyzed 35 papers that met their selection criteria from an initial pool of 24,565 articles. Their goal was to evaluate the current state of integrated NTD control approaches and to identify opportunities for enhancement. The review found that NTD control efforts have been successfully combined with other health initiatives, including water, sanitation, and hygiene programs, vector control, primary healthcare, immunization programs, and malaria management [20]. Furthermore, the results indicated that the integrated management of NTDs is cost-effective and can significantly enhance treatment coverage [20].

A study conducted on Kome Island in Tanzania demonstrated that integrated control strategies are highly effective in reducing schistosomiasis and soil-transmitted helminthiases in both Lake Victoria island and onshore communities [21,22].

Similarly, Knopp and colleagues (2019) explored the impact of integrated interventions on schistosomiasis prevalence through a 5-year repeated cross-sectional cluster-randomized trial [23]. They found that biannual MDA significantly reduced the prevalence and infection intensity of *Schistosoma haematobium* compared to snail control or behavior change activities. However, they emphasized that MDA alone is insufficient to fully interrupt disease transmission, noting that neither snail control nor behavior change activities significantly enhanced the effectiveness of MDA [23].

Numerous studies, along with global and national strategies, have highlighted the need for integrated interventions in NTD control. However, research on the impact of integrated interventions on schistosomiasis prevalence remains limited, and the findings have been inconsistent.

Against this backdrop, this study aimed to assess the effects of an integrated intervention combining school and community-based MDA, community-led total sanitation (CLTS), and citizen voice and action (CVA) on schistosomiasis prevalence in rural Tanzania.

## Methods and materials

### Ethics statement

The study received ethical approval from multiple review boards, including the Joint Institution Review Board of the National Ethical Committee and the Lake Zone Institutional Review Board (MR/53/100/648, Date: 7/8/2020 and MR/53/100/716, Date: 13/10/2022). Before participating, children were given informed consent forms translated into Kiswahili for their parents or guardians to review. Participants aged 9–17 years were provided with an assent form designed to ensure they understood the study's objectives and procedures. Informed consent was obtained in written form from all parents or guardians of all child participants.

All adult participants signed a written informed consent form prior to their involvement in the study.

### Study setting

This study was conducted in the Itilima district of the Simiyu region, located in north-western Tanzania at longitude -3.73333 and latitude 33.48333. As of 2022, the population of Itilima was approximately 412,000, with the Sukuma as the predominant ethnic group [24]. The Itilima district is divided into 22 wards, which are further subdivided into 102 villages and 586 sub-villages(hamlets). Each village comprises 3–5 sub-villages, each with around 100 households, representing the basic unit of administration in Tanzania.

The district experiences annual rainfall ranging from 930 to 1,200 mm, primarily during the long rainy season from January to May. In 2019, 50.4% of the population had access to an improved water supply source, and 38.7% had access to improved sanitation facilities [25]. According to the Strategic Master Plan for the National Neglected Tropical Disease Control Program, this district exhibits a high prevalence of *S. haematobium* infections (nearly 40%) [13]. Prevalence levels vary across the region, and despite repeated MDA, certain villages continue to experience high transmission rates due to persistent environmental vulnerabilities [26].

## Study design

This was a quasi-experimental study involving two rounds of repeated cross-sectional surveys conducted in both the intervention and control arms before and after the intervention. The initial survey took place in September 2020, followed by a second survey in October 2022.

The primary target group of this study was school-age children (SAC). In Tanzania, the age for school-age children ranges from 5 to 17. In this study, the average age of school-aged children was 11.8 (standard deviation: 2.2) years. Out of 98 schools, 12 with a high prevalence of *schistosomiasis* (>10%) in the project area were purposefully selected based on the results of a baseline survey and programmatic considerations [26]. We then assigned school-aged children from these 12 schools to three groups: one control group and two intervention groups. The group with the lowest prevalence of schistosomiasis received only school-based MDA (SMDA) and served as an active control group (group 1). The first intervention group (group 2) underwent both SMDA and community-based MDA (CMDA). The second intervention group (group 3) received an integrated intervention that included SMDA, CMDA, CLTS, and CVA. In this study, adults at the community level were also selected for supplementary investigation. However, we only followed up with adults in 4 out of the 12 sub-villages due to logistical reasons. Figs 1 and 2 illustrates the details of the selection process.

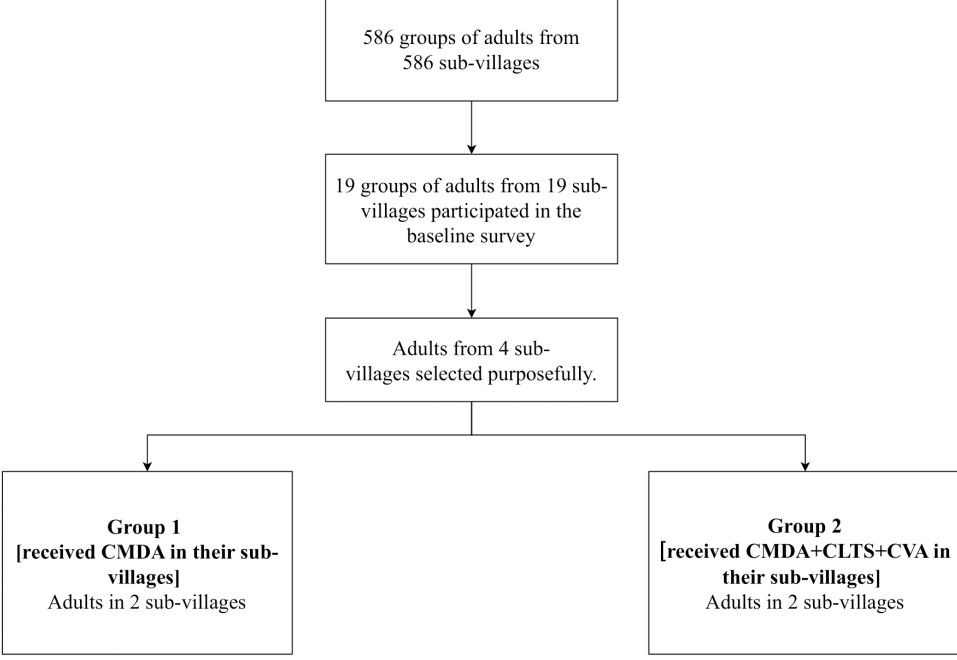

**Fig 1. Flow chart of repeated cross-sectional surveys for school-aged children (trial profile; SMDA: school-based mass drug administration; CMDA: community-based mass drug administration; CLTS: community-based total sanitation; CVA: citizen voice and action).**

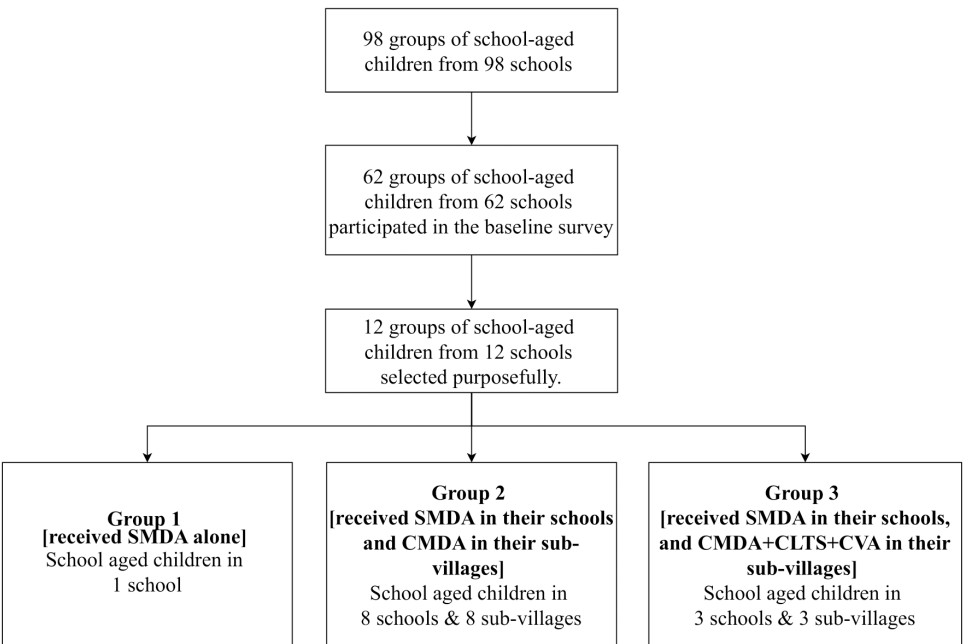

**Fig 2. Flow chart of repeated cross-sectional surveys for adults (trial profile; SMDA: school-based mass drug administration; CMDA: community-based mass drug administration; CLTS: community-based total sanitation; CVA: citizen voice and action).**

## Sampling and sample size

The sample size and sampling strategies were adapted from the study conducted by Cha et al. (2017) and the guide by Hayes & Moulton (2017) [27,28]. A two-stage cluster sampling method was used to select students in schools and residents in sub-villages.

In the first stage, 12 schools with a high prevalence of schistosomiasis were purposively selected. After selecting these schools and accounting for a 16% non-response rate, 60 students—30 boys and 30 girls—were sampled from each institution. A systematic random sampling technique was used to select school-aged children for participation in the study. For supplementary analysis, we tracked adults in 4 sub-villages due to logistical constraints. Therefore, 50 adult community members, comprising 25 men and 25 women, were conveniently selected from the sub-villages surrounding the schools.

## Interventions

### 1) School-based MDA

For SMDA, health teachers, supervised by head teachers, typically manage registration. During SMDA, drugs were distributed across 12 schools in Itilima by school head teachers and health teachers over a period of two days. Parents, informed about the MDA date, prepared food to mitigate severe adverse effects. The first round of the SMDA occurred on March 23–24, 2021, and the second round was conducted on June 7–8, 2022.

### 2) Community-based MDA

CMDA was conducted in 11 sub-villages, comprising eight from group 2 and three from group 3, as planned. Before the CMDA was carried out, community drug distributors (CDDs) received training to perform this task in accordance with the standards and procedures set by the Tanzanian government. In the CMDA process, a team of trained CDDs was

responsible for both the registration and administration of drugs, under the oversight of health officials from nearby dispensaries and health centers. The CDDs conducted household registrations by visiting each one in person. The household registers that were collected were compiled under the supervision of NTD coordinators. The CDDs collaborated with local leaders and health officials within their operational areas. Over a period of 7–9 days, the CDDs visited homes to distribute PZQ to all registered adults and children aged 5–14 who were present at home and had not yet received PZQ during the SMDA. The first round of CMDA took place from October 25 to 31, 2021, and the second round was conducted from June 27 to July 2, 2022.

**3) Community health awareness interventions (CLTS and CVA)**

Community health awareness interventions employing the CLTS and CVA models were applied in three sub-villages. The implementation of CLTS and CVA commenced following the project's kickoff in March 2020. In the first year, we provided cement for people to build toilets, but starting in the second year, we did not provide any additional materials or financial support. The focus was on collective behavior change so that community people could take up the use of household latrines on their own without relying on any external support.

CLTS is an approach designed to achieve open defecation-free (ODF) status by sensitizing community members to the importance of improved hygiene and sanitation, thereby motivating them to voluntarily construct toilets [29,30]. The implementation of CLTS follows a structured process as outlined in the TZA government's CLTS manual, including phases such as pre-triggering, triggering, and post-triggering [31]. In the pre-triggering phase, village leaders and enumerators received training on the CLTS approach and guidelines. They then assessed the presence and types of toilets through household visits, recording their findings in the Health Management Information System database over a period of six days. During the triggering phase, community gatherings were organized to develop a plan for achieving the desired open defection free (ODF) status, and community members began to voluntarily build toilets. The post-triggering stage occurred over the following three months. During this time, local government officials and village leaders made visits to the triggered community once or twice in the first week, and subsequently once every two weeks, although not too frequently, to encourage ongoing toilet construction. Additionally, the team inspected the number of toilets that had been newly constructed or improved, and enumerators collected data over a week during this post-triggering phase.

CVA is a World Vision (WV) model designed to empower community members by providing them with knowledge, information, and skills about specific issues. This enables the community to discuss certain issues, analyze their root causes, and take action [32,33]. CLTS primarily focuses on improving household latrines, whereas CVA aims to enhance latrines in public institutions within the community, such as schools and health centers. Community gatherings, facilitated by leaders, allowed community members to assess their health-related challenges and seek government support and improvements.

**Diagnosis**

A single stool sample was collected from each participant and screened for *S. mansoni* and soil-transmitted helminth eggs using duplicated Kato-Katz thick smears. Each thick smear was prepared with a 41.7 mg template of stool. From each stool sample, four Kato-Katz thick smears were prepared and examined by two independent laboratory technicians, both skilled in the Kato-Katz technique. For quality assurance, a senior laboratory technician, who was blinded to the initial results, re-examined 20% of all positive and negative Kato-Katz thick smears.

A single urine sample was collected from all participants between 10 am and 2 pm in both school and community settings. These samples were first subjected to a gross examination to check for macro-hematuria. Additionally, a urine dipstick/urinalysis reagent strip (Mission, Expert, USA) was used to test for micro-hematuria. For further analysis, the samples underwent a urine filtration process, and the filters were examined under a light microscope to detect the presence of *S. haematobium* eggs. To maintain quality control, a senior laboratory technician re-examined 20% of the samples, including both positive and negative results, before the end of the field day.

## Survey questionnaire

Face-to-face interviews were conducted with selected school-aged children (SACs) and adults using a pre-tested questionnaire. The structured questionnaire, initially developed in English, was translated into Kiswahili. The accuracy of this translation was verified during enumerator training, where all questions and response options were thoroughly discussed to ensure a common understanding among team members. While both the SACs' and adults' questionnaires included questions about household demographics, only the SACs' questionnaire contained questions regarding defecation and hygiene behaviors, as well as toilet conditions.

## Primary outcome

The primary outcome of this study was the prevalence of schistosomiasis in individuals infected with *S. haematobium* or *S. mansoni*. We defined positivity for schistosomiasis as a positive test result from either the Kato-Katz or urine filtration method. In the study area, the prevalence of schistosomiasis with *S. haematobium* was predominant (10.1%) among schoolchildren, whereas the prevalence of *S. mansoni* was minimal (0.3%) at baseline in 2020.

## Intermediate outcomes

We used water contact behavior, experience of taking praziquantel, the presence of a latrine at home, the presence of a latrine at school, the use of a latrine at home, and hand washing behaviors, which were measured based on household survey using questionnaire. Water contact behavior was defined as coming into contact with surface water such as river, streams, ponds, canal for swimming, fetching water, doing laundry, fishing, watering livestock at least 3 times per week.

## Statistical analysis

The data were double-entered into a Microsoft Excel sheet, cleaned, and then exported to R version 4.3.1. The primary objective of this analysis was to assess the impact of integrated interventions on the prevalence of *Schistosoma haematobium*.

Multilevel logistic regression models were used to evaluate the impact of integrated interventions on the prevalence of schistosomiasis, with individuals at level one and groups (schools or sub-villages) at level two. Risk differences were calculated. Changes in proportions were considered to be significant if the 95% confidence intervals (CIs) did not overlap. The findings were reported as both crude and adjusted risk differences, with adjustments made for sociodemographic indicators, including gender and age.

# Results

## Basic characteristics

This study included 1,457 students from 12 schools, consisting of 716 boys (49.1%) and 741 girls (50.9%). The students' average age was 11.8 ± 2.2 years. Table 1 displays the general characteristics of the students.

A total of 397 adults from four sub-villages participated in the study, representing 16,712 beneficiaries across 3,343 households. Of these participants, 217 (54.7%) were male, and 180 (45.3%) were female. The average age of the participants was 25.8 years, with a standard deviation of 14.9 years. Detailed general characteristics of the adults are presented in Table 2.

## Results for primary outcomes

Baseline and endline values of schistosomiasis prevalence are illustrated in Fig 3.

The SMDA-only group exhibited a lower prevalence of schistosomiasis infection than both the SMDA plus CMDA group (11.5% vs. 27.3%; odds ratio [OR] 1.062, p < .05) and the fully integrated model group (11.5% vs. 37.6%; OR 1.535;

**Table 1. Basic characteristics of students at baseline and endline.**

| | SMDA only | | SMDA plus CMDA | | Fully integrated model | |
|---|---|---|---|---|---|---|
| | **Baseline** | **Endline** | **Baseline** | **Endline** | **Baseline** | **Endline** |
| Schools | 1 | | 8 | | 3 | |
| Total beneficiaries (2020) | 1,184 | | 10,341 | | 3,553 | |
| Participants with outcome data | 61 | 60 | 495 | 480 | 181 | 180 |
| Mean age in years (SD) | 11.180 (2.240) | 12.267 (2.162) | 11.360 (2.648) | 12.131 (1.758) | 11.663 (2.264) | 12.250 (1.864) |
| Gender | | | | | | |
| Male | 30(49.2%) | 29(48.3%) | 254(51.3%) | 226(47.1%) | 85(47.0%) | 92(51.1%) |
| Female | 31(50.8%) | 31(51.7%) | 241(48.7%) | 254(52.9%) | 96(53.0%) | 88(48.9%) |
| Mean number of households (SD) | 15.934 (5.147) | 10.914 (4.907) | 7.885 (3.394) | 9.530 (4.252) | 9.717 (4.159) | 9.137 (3.913) |
| Education level of mother* | (N=61) | (N=60) | (N=427) | (N=473) | (N=180) | (N=169) |
| 0 | 7(11.48%) | 11(18.33%) | 27(6.32%) | 79(17.06%) | 64(35.56%) | 22(13.02%) |
| 1 | 14(22.95%) | 39(65.00%) | 72(16.86%) | 42(9.07%) | 14(7.78%) | 21(12.42%) |
| 2 | 40(65.57%) | 7(11.67%) | 324(75.88%) | 305(65.87%) | 101(56.11%) | 98(57.99%) |
| 3 | 0(0.00%) | 3(5.00%) | 4(0.94%) | 37(7.99%) | 1(0.56%) | 28(16.57%) |
| Education level of father* | (N=61) | (N=59) | (N=427) | (N=456) | (N=180) | (N=167) |
| 0 | 2(3.28%) | 6(10.17%) | 15(3.51%) | 45(9.87%) | 33(18.13%) | 8(4.79%) |
| 1 | 8(13.11%) | 43(72.88%) | 60(14.05%) | 52(11.40%) | 11(6.11%) | 32(19.16%) |
| 2 | 51(83.61%) | 8(13.56%) | 313(73.30%) | 308(67.54%) | 134(74.44%) | 95(56.89%) |
| 3 | 0(0.00%) | 2(3.39%) | 39(9.13%) | 51(11.18%) | 2(1.11%) | 32(19.16%) |
| Occupation of mother** | (N=60) | (N=58) | (N=403) | (N=468) | (N=169) | (N=178) |
| a | 57(95.0%) | 56(96.6%) | 356(88.3%) | 393(84.0%) | 161(95.3%) | 128(71.9%) |
| b | 3(5.0%) | 0(0.0%) | 40(9.9%) | 19(4.1%) | 7(4.1%) | 19(10.7%) |
| c | 0(0.0%) | 0(0.0%) | 4(1.0%) | 43(9.2%) | 0(0.0%) | 21(11.8%) |
| d | 0(0.0%) | 2(3.4%) | 3(0.8%) | 13(2.8%) | 1(0.6%) | 10(5.6%) |
| Occupation of father** | (N=51) | (N=60) | (N=357) | (N=469) | (N=166) | (N=186) |
| a | 44(86.3%) | 52(86.6%) | 287(80.4%) | 393(83.8%) | 158(95.2%) | 140(75.3%) |
| b | 2(3.9%) | 0(0.0%) | 14(3.9%) | 2(0.4%) | 5(3.0%) | 3(1.6%) |
| c | 4(7.8%) | 4(6.7%) | 38(10.6%) | 45(9.6%) | 1(0.6%) | 17(9.1%) |
| d | 1(2.0%) | 4(6.7%) | 18(5.1%) | 29(6.2%) | 2(1.2%) | 26(14.0%) |

*0: Illiterate/did not go to school.

1: Did not complete primary school.

2: Completed primary school.

3: Over primary school.

**multiple response.

a: Farmer.

b: Housekeeper.

c: Small business.

d: Employed.

p<.001) at baseline (Table 3). However, by the endline, the prevalence of schistosomiasis infections was low across all intervention groups, with rates of 0.0%, 2.1%, and 2.8%.

The effects of interventions on schistosomiasis prevalence among students are presented in Table 3 and Fig 4. Examining these effects, the SMDA plus CMDA group (risk difference [RD] -13.7%; 95% CI -24.5% to -2.9%; p<.05) and the

**Table 2. Basic characteristics of adults at baseline and endline.**

| | SMDA plus CMDA | | Fully integrated model | |
|---|---|---|---|---|
| | **Baseline** | **Endline** | **Baseline** | **Endline** |
| Sub-villages | 2 | | 2 | |
| Total households (2019) | 2,260 | | 1,083 | |
| Total beneficiaries (2019) | 11,298 | | 5,414 | |
| Participants with outcome data | 105 | 100 | 92 | 100 |
| Mean age in years (SD) | 19.69 (9.89) | 28.72 (14.39) | 25.09 (16.53) | 29.86 (16.25) |
| Gender | | | | |
| Male | 54(51.4%) | 67(67.0%) | 40(43.5%) | 56(56.0%) |
| Female | 51(48.6%) | 33(33.0%) | 52(56.5%) | 44(44.0%) |

**Fig 3. Baseline and endline prevalence of schistosomiasis in school-aged children (0: the group that received school-based mass drug administration alone; 1: the group that received school-based mass drug administration in their school and community-based mass drug administration in their community; 123: the group that received school-based mass drug administration in their school, and community-based mass drug administration, community-led total sanitation and citizen voice and action in their community.** There was no school with a prevalence between 40–49%. Source of the basemap shapefile: https://data.humdata.org/dataset/cod-ab-tza (UN OCHA)).

fully integrated model group (RD -23.3%; 95% CI -32.7% to -13.9%; p<.001) both showed a significantly greater reduction in schistosomiasis infection compared to the SMDA-only group. Additionally, the fully integrated model group (RD -9.6%; 95% CI -23.3% to 4.1%; p>.05) exhibited a greater reduction in infection rates than the SMDA plus CMDA group, although this difference did not achieve 95% statistical significance.

Table 4 presents the impact of various interventions on the prevalence of schistosomiasis among adults. There were significant differences in the reduction of schistosomiasis infections among adults when comparing the SMDA plus CMDA group with the fully integrated model group. The fully integrated model group demonstrated a significantly greater

**Table 3. Effects of interventions on schistosomiasis prevalence (SAC).**

| Survey round | | SMDA only (active control) Baseline | Endline | SMDA plus CMDA Baseline | Endline | Fully integrated model (SMDA, CMDA, CLTS, CVA) Baseline | Endline | |
|---|---|---|---|---|---|---|---|---|
| n/N (prevalence, %) | | 7/61 (11.5%) | 0/60 (0.0%) | 135/495 (27.3%) | 10/480 (2.1%) | 68/181 (37.6%) | 5/180 (2.8%) | |
| OR (95% CI) | | 1 | 1 | 1.062* (0.25, 1.874) | 15.716 (-2705.4, 2736.8) | 1.535*** (0.692, 2.378) | 17.011 (-4469.3, 4503.3) | |
| Risk difference (95% CI) | Ref. SMDA | | | 15.8%*** (6.9%, 24.7%) | NA | 26.1%*** (15.4%, 36.8%) | NA | |
| | Ref. Baseline | | -11.5%** (-19.5%, -3.5%) | | -25.2%*** (-29.3%, -21.1%) | | -34.8%***(-42.2%, -27.3%) | |
| | DiD | | | | -13.7%* (-24.5%, -2.9%) | | -23.3%*** (-32.7%, -13.9%) | -9.6% (-23.3%, 4.1%) |
| | Adjusted DiD (only gender) | | | | NA | | NA | -9.5% (-23.3%, 4.3%) |

*Abbreviations: OR odds ratio, CI confidence interval, NA not applicable, SMDA school mass drug administration, CMDA community mass drug administration, DiD difference-in-difference, CLTS community-led total sanitation, CVA citizen voice and action

*p-value: p<.05 *, p<.01 **, p<.001 ***

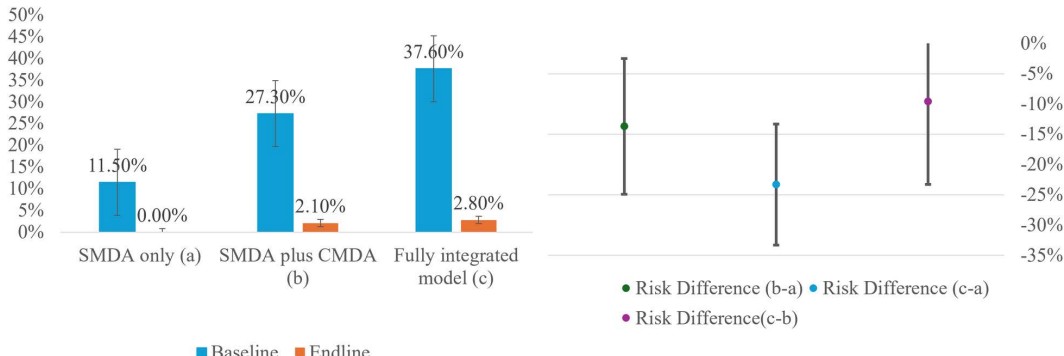

**Fig 4. Baseline and endline values of schistosomiasis prevalence (left) and the effect of intervention (difference-in-difference) (SMDA: school-based mass drug administration; CMDA: community-based mass drug administration; Fully integrated model: SMDA+ CMDA+ Community-based total sanitation + citizen voice and action).**

reduction in schistosomiasis infection rates compared to the SMDA plus CMDA group (RD -39.6%; 95% CI -56.4% to -22.8%, p<.001).

The SMDA plus CMDA group exhibited a lower baseline prevalence of schistosomiasis infection compared to the fully integrated model group (10.5% vs. 51.1%, OR = 2.189, p<.001). However, by the end of the study, both groups demonstrated low prevalence rates, with schistosomiasis infections at 1.0% and 2.0%, respectively.

## Effects on intermediary outcomes

Table 5 outlines the changes in defecation and hygiene behaviors among students, as reported in the school-based questionnaire across the three intervention arms.

**Table 4. Effects of interventions on schistosomiasis prevalence (adults).**

| | | SMDA plus CMDA (active control) | | Fully integrated model (SMDA, CMDA, CLTS, CVA) | |
|---|---|---|---|---|---|
| Survey round | | Baseline | Endline | Baseline | Endline |
| n/N (prevalence, %) | | 11/105 (10.5%) | 1/100 (1.0%) | 47/92 (51.1%) | 2/100 (2.0%) |
| OR (95% CI) | | 1 | 1 | 2.189*** (1.442, 2.935) | 0.703 (-1.713, 3.120) |
| Risk difference (95% CI) | Ref. CMDA | | | 40.6%*** (28.8%, 52.4%) | 1.0% (-2.4%, 4.4%) |
| | Ref. Baseline | | -9.5%** (-15.6%, -3.3%) | | -49.1%*** (-59.7%, -38.5%) |
| | DiD | | | | -39.6%*** (-56.4%, -22.8%) |
| | Adjusted DiD (gender, age) | | | | NA |

*Abbreviations: OR odds ratio, CI confidence interval, NA not applicable, SMDA school mass drug administration, CMDA community mass drug administration, DiD difference-in-difference, CLTS community-led total sanitation, CVA citizen voice and action

*p-value: p<.05 *, p<.01 **, p<.001 ***.

Initially, the group combining SMDA and CMDA had the lowest frequency of water contact behavior exceeding three times daily among the three groups, with the SMDA-only group next, followed by the fully integrated model group. By the end of the study, however, this pattern reversed. The SMDA plus CMDA group showed the highest frequency of such behavior, followed by the fully integrated model group and then the SMDA-only group. Throughout the project, there was an increase in the frequency of water contact behavior more than three times a day in the SMDA plus CMDA group, whereas the other two groups saw a decrease. When comparing the changes in proportions, the fully integrated model group exhibited a significant reduction in water contact behavior compared to the SMDA plus CMDA group, with a relative difference of -56.0% and a confidence interval ranging from -67.3% to -44.8% (p<.05).

The percentage of students reporting improved home toilets increased across all three groups, with the most significant rise observed in the fully integrated model group. This group was followed by the SMDA plus CMDA group, and then the SMDA-only group. Handwashing after using the latrine showed the greatest increase in the fully integrated model group. There was a moderate increase in the SMDA plus CMDA group, whereas a decrease was noted in the SMDA-only group. A comparative analysis of the changes among the groups indicated that the increase in the fully integrated model group was significantly greater than that in the SMDA-only group (RD 48.0%; 95% CI 5.3% to 90.7%, p<.05). However, the difference was not statistically significant for the SMDA plus CMDA group (RD 31.5%; 95% CI -19.7% to 82.7%, p>.05).

The initial proportion of home toilet use was over 90% in all three groups at baseline but declined in each group by the end of the period. This decrease was smallest in the fully integrated model group, followed by the SMDA-only group, and then the SMDA plus CMDA group. When comparing the extent of change among the groups, the fully integrated model group demonstrated significantly better outcomes than the other two groups.

The separate analysis for boys and girls showed significant differences between the integrated model and the CMDA model in water contact behavior for both genders (S1 Table and S3 Table). Positive changes were identified under the integrated model, compared to the CMDA model, for both boys and girls in behaviors such as using the latrine at home for defecation and washing hands after using the latrine. However, distinct differences in prevalence between the intervention groups were found according to gender. Among boys, a significant difference was noted between the integrated model and the CMDA model (S2 Table), while for girls, a significant difference was observed between the SMDA model and the CMDA model (S4 Table). Notably, the integrated model, which combined factors related to behavioral change, demonstrated a more pronounced effect on boys than on girls.

**Table 5. Effects on intermediate outcomes among SACs.**

| Survey round | SMDA only (active control) | | SMDA plus CMDA | | Fully integrated model (SMDA, CMDA, CLTS, CVA) | | Ref. SMDA | Ref. CMDA |
|---|---|---|---|---|---|---|---|---|
| | Baseline | Endline | Baseline | Endline | Baseline | Endline | | |
| Water contact behavior (3 times or more) n/N (%) | 44/61 (72.1%) | 14/60 (23.3%) | 174/427 (40.7%) | 295/465 (63.4%) | 146/180 (81.1%) | 86/180 (47.8%) | 15.5% (-12.3%,43.2%) | -56.0%*(-67.3%,-44.8%) |
| Praziquantel[1] | 60/61 (98.4%) | 56/58 (96.6%) | 277/427 (64.9%) | 419/472 (88.8%) | 120/180 (66.7%) | 162/169 (95.9%) | 31.0% (-1.1%,63.1%) | 5.3% (-31.5%,42.1%) |
| Latrine at home | No (0/61, 0%) | No (0/60, 0%) | No (4/495, 0.8%) | No (0/480, 0%) | No (0/181, 0%) | No (0/180, 0%) | NA | 10.0% (-11.9%,31.9%) |
| | Unimproved (60/61, 98.4%) | Unimproved (50/60, 83.3%) | Unimproved (379/495, 76.6%) | Unimproved (328/480, 68.3%) | Unimproved (172/181, 95.0%) | Unimproved (110/180, 61.1%) | | |
| | Improved (0/61, 0%) | Improved (3/60, 5.0%) | Improved (7/495, 1.4%) | Improved (123/480, 25.6%) | Improved (1/181, 0.6%) | Improved (62/180, 34.4%) | | |
| Latrine at school | 51/61 (83.6%) | 60/60 (100%) | 376/427 (88.1%) | 480/480 (100%) | 147/180 (81.7%) | 180/180 (100%) | 1.9% (-26.0%,29.9%) | NA |
| Use latrine at home to defecate & urinate | 60/61 (98.4%) | 44/60 (73.3%) | 391/427 (91.6%) | 309/479 (64.5%) | 171/180 (95.0%) | 170/180 (94.4%) | 25.0%*** (23.2%, 26.8%) | 27.0%* (2.5%, 51.6%) |
| Wash hands after using latrine | 56/61 (91.8%) | 47/58 (81.0%) | 218/427 (51.1%) | 268/472 (56.8%) | 97/180 (53.9%) | 164/180 (91.1%) | 48.0%* (5.3%, 90.7%) | 31.5% (-19.7%,82.7%) |
| Wash hands before having meals | 61/61 (100%) | 58/58 (100%) | 418/427 (97.9%) | 474/477 (99.4%) | 179/180 (99.4%) | 179/180 (99.4%) | NA | -1.5% (-5.6%, 2.6%) |

*Abbreviations: NA not applicable.

*p-value: p<.05 *, p<.01 **, p<.001 ***.

[1]Having taken praziquantel in the past.

## Discussion

The results indicate a more pronounced potential effect of integrated NTD control measures, which include MDA along with CLTS and health education, compared to relying solely on SMDA. We observed that the prevalence of schistosomiasis in the SMDA plus CMDA group and the fully integrated group was not 0%. First, an explanation for this might relate to differences in the praziquantel compliance rate per group. Second, this study involved a repeated cross-sectional survey; hence, we could not ensure that the participants at the endline had received all the interventions according to the plan. Third, a review [34] reported that comparable cure rates of praziquantel were 81.2%–99.1% for intestinal and 79%–93.7% for urinary schistosomiasis studies. As we measured the infection status at least a few months after MDA, we believe that some individuals remained positive even after taking praziquantel.

For students, the combination of SMDA and CMDA, as well as the integrated model, proved more effective than SMDA alone. There was a substantial decline in water contact behavior, along with increased coverage of improved latrines and treatment in the fully integrated model. Similarly, the SMDA plus CMDA group saw increased coverage of improved latrines and treatment with PZQ. These factors likely contributed to the significant reduction in schistosomiasis observed in the two control groups. This underscores the need for a comprehensive approach that targets out-of-school youth, preschoolers, adults, and other demographics. Additionally, among adults, the integrated model surpassed CMDA alone,

demonstrating the effectiveness of combining hygiene facilities and behavioral improvements through CLTS and CVA with MDA.

Furthermore, these intervention components not only contributed to reducing *S. haematobium* but also directly improved students' hygiene behaviors, including water contact behavior and toilet use, as intermediate achievements [35].

An important aspect of this study is its demonstration of the effects of adult-centered activities, such as CLTS and CVA, on hygiene and behavioral changes. These interventions not only impacted adults, but also affected students. This underscores the broad impact and interconnectedness of interventions targeting various demographic groups within the community. Overall, the findings highlight the holistic nature of the interventions implemented in the study, which not only aimed to reduce schistosomiasis prevalence but also addressed broader health and hygiene behaviors among both adults and students. Integrated approaches of this type are crucial for achieving sustainable improvements in health outcomes within communities.

## Implications of this study

Despite the growing acknowledgment of the need for integrated NTD interventions to support MDA, enhance WASH, and offer health education against schistosomiasis, evidence of the effectiveness of integrated NTD control within Tanzania remains scarce. Furthermore, recent randomized controlled trials exploring the impact of integrated NTD control on the prevalence of NTD diseases have produced mixed results [23,35,36]. Therefore, this study makes a significant contribution to addressing this gap by showing that the integrated approach to eradicating *S. haematobium* was consistently effective among both students and adults.

Previous studies on the effectiveness of integrated NTD projects, such as the one conducted in Côte d'Ivoire [35], have focused on assessing the prevalence of various NTDs among 810 students and adults, including *S. haematobium*, soil-transmitted helminths, and hookworm. Although the integrated project did not demonstrate significant effects on *S. haematobium*, it did observe notable reductions in the prevalence of soil-transmitted helminths and hookworms. Similarly, Knopp and his colleagues (2019) sought to evaluate the effectiveness of an integrated approach that combined MDA, behavioral change education, and snail control, specifically targeting *S. haematobium* [23]. However, their study was limited by its focus solely on student populations, failing to capture the broader community impact of the interventions. In contrast, our study incorporated CLTS to improve sanitation facilities alongside behavioral change, offering a more holistic approach. This result aligns with studies indicating that large-scale integrated approaches to controlling neglected tropical diseases—including chemotherapy, snail control, and health education—have significantly contributed to reducing prevalence rates in China and Thailand [37–39].

Furthermore, this study is consistent with existing literature that confirms the effectiveness of CLTS [29,30,40]. Research conducted in Ghana has shown that CLTS projects tend to be more effective in small, remote villages, where community mobilization is facilitated by strong cohesion and a clear hierarchical structure [30]. Considering that our project focused on community participation in remote areas, similar factors likely influenced our results.

## Policy implications of the study

First, Tanzania should continue to maintain and strengthen its SMDA system for NTD prevention, as evidenced by the reduction in *S. haematobium* prevalence from 11.5% to 0.0% among SMDA-only groups in this study. Since the launch of the new Schistosomiasis Control Program in Unguja in 2003 [15] Zanzibar successfully achieved countrywide coverage for all school children using praziquantel from 2004 to 2006. In 2004, the National Schistosomiasis and Soil-Transmitted Helminth Control Program was established under the Ministry of Health and Social Welfare with support from SCI on the mainland of Tanzania. According to the NSSCP report, the School-based Mass Deworming Treatment has covered 21 regions. Since then, the program has implemented SMDA, where all primary school children receive praziquantel

regardless of their infection status, as a blanket mass treatment strategy. Given the long-standing success of the SMDA system in Tanzania, it is crucial to continue supporting and enhancing this system. Additionally, integrating behavioral change education into school curricula could significantly bolster comprehensive NTD prevention efforts.

Second, by emphasizing SMDA and concurrently implementing CMDA, we can extend preventive measures to include adults who interact with students and preschoolers. According to the NSSCP, efforts to control schistosomiasis and soil-transmitted helminthiases in Tanzania initially targeted the most at-risk age groups, specifically elementary school students aged 5–14 years [11,13]. While SMDA is cost-effective, it alone is insufficient to fully address the issue [41]. However, according to Tanzania's strategic master plan for NTD [13], MDA for soil-transmitted helminths is done at the community and school level, whereas MDA for schistosomiasis is performed in schools and thus needs to be supplemented. Achieving the goals outlined in the NTD roadmap [9,12,13] requires a combination of SMDA and CMDA centered on the community. The failure of the policy in the Philippines to effectively control NTDs highlights the critical role of sufficient political commitment, adequate drug coverage and compliance, and the free availability of high-quality praziquantel in the successful implementation of MDAs for combating schistosomiasis [37,42].

Third, concurrent improvements in WASH activities, such as CLTS, are essential for approaching a fundamental solution to NTDs. Tanzania has already emphasized the importance of basic water supply, sanitation, and hygiene in NTD-endemic areas within its strategic master plan and the National Sanitation Campaign has introduced CLTS as a means of behavioral change communication for this purpose [13]. To make these efforts more effective, governments should work with various community-based organizations to ensure that communities are well mobilized [12].

Fourth, as prevalence gradually decreases, it is advisable to utilize primary healthcare platforms for diagnosis and administration throughout the year, rather than relying solely on periodic MDAs. In this regard, the integration of NTDs into health system structures within Tanzania's sustainability plan is relevant and should be further strengthened [12].

Lastly, behavioral change education, when combined with community participation, has been shown to significantly influence knowledge and hygiene behaviors associated with schistosomiasis [22,43]. Integrated NTD control efforts in northeastern Thailand, particularly through the Lawa model, also have demonstrated that a holistic, bottom-up approach—referred to as One Health or EcoHealth—can effectively disrupt disease transmission [39]. This approach emphasizes monitoring the entire ecosystem and relies heavily on active community participation. Therefore, interventions aimed at addressing knowledge, attitudes, and behaviors concerning NTDs should emphasize community engagement and systematic planning.

The limitations and strengths of this study are as follows: The study was initially flawed due to a lack of a structured design, and there were uncontrolled spill-over effects between treatment groups, which impacted the evaluation of CLTS and CVA activities. Being an implementation study, it demonstrated disparities in initial values among project groups, which may have been influenced by external factors. Further research is necessary to determine the true impact of the intervention. Non-random sampling in community-based tests may introduce selection bias, unlike school-based tests that employ random sampling. Furthermore, surveys conducted exclusively among students restrict the direct assessment of behavioral changes in adults involved in CLTS and CVA activities. Given the relatively small scale of the study and its duration of less than 3 years, caution should be exercised when generalizing the findings to larger-scale or longer-term projects. Further validation is necessary to confirm these results. Despite these limitations, this study offers valuable insights into the effectiveness of NTD control projects, especially regarding the incremental integration of intervention components.

Randomized controlled trials are the acknowledged standard in evaluating effectiveness and are recognized as best method for generating evidence [44]. Randomly allocating participants can ensure the even distribution of confounding factors between the intervention and control groups, which minimizes bias in assessing the effects of an intervention. We had to rely on quasi-experimental design for this study. Therefore, we cannot rule out selection bias, since an even distribution of confounding factors between the comparison groups were not ensured, which affects the robustness of this study.

We propose a follow-up study with the following objectives: First, to confirm the effectiveness of the integrated project on a larger scale, more robust methods such as randomized control trials should be employed. This approach will provide stronger evidence of the impact of the integrated strategy on NTD control. Second, to verify the effectiveness of strengthening primary care, which includes focal distribution and MDA, it is essential to evaluate the impact of these interventions on the prevalence and incidence rates of NTDs. This evaluation will help assess the feasibility and effectiveness of integrating NTD control into primary healthcare systems. Third, evaluating the effectiveness of snail control as part of the integrated project is essential. Snail control plays a vital role in NTD control strategies, but this study did not address its effectiveness. Thus, assessing the impact of snail control interventions on NTD transmission rates will offer valuable insights into the overall success of integrated NTD control efforts. To enhance the accuracy and precision of measurements concerning NTD prevalence and infection rates, it is crucial to employ more sensitive diagnostic techniques, such as circulating cathodic antigen tests. These methods facilitate a more comprehensive assessment of the impact of interventions on disease transmission and control. By addressing these objectives in a follow-up study, we can achieve a deeper understanding of the effectiveness of integrated NTD control strategies and identify areas for improvement in NTD prevention and control efforts.

## Conclusion

Integrated NTD control approaches, which include MDA, CLTS, CVA, and health education, have proven effective in reducing the prevalence of infectious diseases and enhancing community health outcomes. Our study, conducted in the rural Itilima District where prevalences range from 5% to 10%, provides a replicable model for other communities facing similar health challenges. Sustained implementation of these comprehensive interventions is crucial for overcoming the ongoing obstacles in NTD control programs, such as limited resources, local resistance, and issues with compliance. By scaling up these integrated NTD interventions, communities can significantly advance toward meeting Sustainable Development Goal 3.3, which focuses on ending communicable diseases and alleviating the global burden of NTDs.

Overall, the success of integrated NTD control approaches in Itilima District highlights their potential for broad application across diverse communities, paving the way for enhanced health outcomes and sustainable disease control efforts.

## Supporting information

**S1 Text. Questionnaire set.**
(DOCX)

**S1 Table. Effects on intermediate outcomes among male school-aged children.**
(DOCX)

**S2 Table. Effects of interventions on schistosomiasis prevalence (male school-aged children).**
(DOCX)

**S3 Table. Effects on intermediate outcomes among female school-aged children.**
(DOCX)

**S4 Table. Effects of interventions on schistosomiasis prevalence (female school-aged children).**
(DOCX)

**S1 Data. Baseline and Endline Data for adults.**
(XLSX)

**S2 Data. Baseline and Endline Data for students.**
(XLSX)

## Acknowledgments

We are grateful to the WV Tanzania Itilima NTD elimination project team, Wold Vision Tanzania Head Office, and the World Vision Korea for their administrative and technical support. We extend our thanks to the community people and students in the target areas for their participation in this study.

## Author contributions

**Conceptualization:** Yoonho Cho, Jungim Lee, Seungman Cha.

**Data curation:** Yoonho Cho, Humphrey Deogratias Mazigo, Seungman Cha.

**Formal analysis:** Yoonho Cho, Seungman Cha.

**Funding acquisition:** Jungim Lee.

**Investigation:** Jungim Lee, Humphrey Deogratias Mazigo, Leah Elisha Salamba, Seungman Cha.

**Methodology:** Yoonho Cho, Jungim Lee, Seungman Cha.

**Project administration:** Jungim Lee, Leah Elisha Salamba.

**Resources:** Jungim Lee, Humphrey Deogratias Mazigo, Leah Elisha Salamba.

**Software:** Yoonho Cho, Seungman Cha.

**Supervision:** Jungim Lee, Humphrey Deogratias Mazigo, Seungman Cha.

**Validation:** Humphrey Deogratias Mazigo, Seungman Cha.

**Visualization:** Yoonho Cho, Seungman Cha.

**Writing – original draft:** Yoonho Cho, Seungman Cha.

**Writing – review & editing:** Jungim Lee, Humphrey Deogratias Mazigo, Leah Elisha Salamba.

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
