## [Decision Letter · Decision Letter 0]

PNTD-D-24-01456

Effects of an integrated intervention on schistosomiasis prevalence in a rural area of Tanzania

Dear Dr. Cha,

Thank you for submitting your manuscript to PLOS Neglected Tropical Diseases. After careful consideration, we feel that it has merit but does not fully meet PLOS Neglected Tropical Diseases's publication criteria as it currently stands. Therefore, we invite you to submit a revised version of the manuscript that addresses the points raised during the review process.

Please submit your revised manuscript within 60 days Feb 14 2025 11:59PM. If you will need more time than this to complete your revisions, please reply to this message or contact the journal office at plosntds@plos.org. Please include the following items when submitting your revised manuscript:

We look forward to receiving your revised manuscript.

Kind regards,

Sutas Suttiprapa, Ph.D.

Academic Editor

Krystyna Cwiklinski

Section Editor

Shaden Kamhawi

co-Editor-in-Chief

Paul Brindley

co-Editor-in-Chief

**Additional Editor Comments :**

Dear Dr. Seungman Cha and co-authors,

Thank you for submitting this important and valuable manuscript assessing the effectiveness of integrated control efforts against S. mansoni and S. haematobium in an endemic region in Tanzania. The study provides strong evidence supporting the benefits of integrated approaches to schistosomiasis control, which are highly relevant for policymakers and public health practitioners. However, the manuscript requires significant revisions to improve clarity, quality, and general interpretability of the findings. Below are detailed comments and suggestions for improvement:

**Journal Requirements:**

At this stage, the following Authors/Authors require contributions: Yoonho Cho, Jungim Lee, Humphrey Mazigo, Leah Elisha Salamba, and Seungman Cha. Please ensure that the full contributions of each author are acknowledged in the "Add/Edit/Remove Authors" section of our submission form.

- ® on page: 8.

3) Thank you for including an Ethics Statement for your study. Please include:

i) A statement that formal consent was obtained (must state whether verbal/written) from the parent/guardian. of the child participants.

5) We have noticed that you have uploaded Supporting Information files, but you have not included a list of legends. Please add a full list of legends for your Supporting Information files after the references list.

7) Your current Financial Disclosure states, "This research was supported by the Korea International Cooperation Agency (KOICA) and World Vision Korea under the title of "Neglected Tropical Diseases Elimination Project in Itilima District, Tanzania” between 2020 and 2022. (Letter of Grant Agreement for KOICA's Public-Private Partnership Projects with Global Disease Eradication Fund No.2020-01) ".

However, your funding information on the submission form indicates one funder. Please ensure that the funders and grant numbers match between the Financial Disclosure field and the Funding Information tab in your submission form. Note that the funders must be provided in the same order in both places as well.                                                  . 

Please indicate by return email the full and correct funding information for your study and confirm the order in which funding contributions should appear. Please be sure to indicate whether the funders played any role in the study design, data collection and analysis, decision to publish, or preparation of the manuscript.

8) We noticed that the Data Availability Statement mentioned in the manuscript is different from that provided in the online submission form. Please ensure that the Data Availability statement is the same in both places.

**Comments to the Authors:**

**Please note that the reviews are uploaded as attachments.**

**Reviewers' Comments:**

Reviewer's Responses to Questions

**Key Review Criteria Required for Acceptance?**

**Methods**

-Are the objectives of the study clearly articulated with a clear testable hypothesis stated?

-Is the study design appropriate to address the stated objectives?

-Is the population clearly described and appropriate for the hypothesis being tested?

-Is the sample size sufficient to ensure adequate power to address the hypothesis being tested?

-Were correct statistical analysis used to support conclusions?

-Are there concerns about ethical or regulatory requirements being met?

Reviewer #1: Please see attached file.

Reviewer #2: The study was well designed.

Reviewer #3: (No Response)

**Results**

-Does the analysis presented match the analysis plan?

-Are the results clearly and completely presented?

-Are the figures (Tables, Images) of sufficient quality for clarity?

Reviewer #1: Please see attached file.

Reviewer #2: The results were presented well.

Reviewer #3: (No Response)

**Conclusions**

-Are the conclusions supported by the data presented?

-Are the limitations of analysis clearly described?

-Do the authors discuss how these data can be helpful to advance our understanding of the topic under study?

-Is public health relevance addressed?

Reviewer #1: Please see attached file.

Reviewer #2: Conclusions were made based on the study findings.

Reviewer #3: (No Response)

**Editorial and Data Presentation Modifications?**

Reviewer #1: (No Response)

Reviewer #2: The manuscript should be revised before acceptance.

Reviewer #3: (No Response)

**Summary and General Comments**

Reviewer #1: Please see attached file.

Reviewer #2: The authors deserve praise for starting the study during the COVID-19 outbreak and seeing it through to completion. Particularly in rural Tanzania, I can only image the logistical difficulties and perseverance required to complete the study. Nonetheless, I have several comments for the authors below:

Were there any lakes near the study sites?

What is the age range of school-age children?

Line 133 - Is there a reference for the baseline survey?

Table 1 - What is the meaning of peasant? Would it be more appropriate to categorize it as unemployed or having odd-jobs, instead?

Table 1 - Suggest to revise Housewife/Househusband or other suitable term to describe the housewife-equivalent for the husband.

Line 193 - ODF should be described in full at the first mention.

For the CLTS and the CVA activities, was funding provided to the study participants for the improvement of toilets?

What do the authors mean by water contact behavior?

Table 4 - what does Praziquantel mean?

Can the authors explain why Endline Prevalences for SMDA plus CMDA and the fully integrated group does not show 0%, similar to the control group, since MDA was administrated across all groups? It should show 0% post-treatment, unless there are other explanations. This should also be discussed.

Reviewer #3: (No Response)

PLOS authors have the option to publish the peer review history of their article (what does this mean? ). If published, this will include your full peer review and any attached files.

**Do you want your identity to be public for this peer review?** For information about this choice, including consent withdrawal, please see our Privacy Policy .

Reviewer #1: No

Reviewer #2: No

Reviewer #3: No

**Figure resubmission:**
---

## [Decision Letter · Decision Letter 1]

PNTD-D-24-01456R1Effects of an integrated intervention on schistosomiasis prevalence in a rural area of TanzaniaPLOS Neglected Tropical Diseases Dear Dr. Cha, Thank you for submitting your manuscript to PLOS Neglected Tropical Diseases. After careful consideration, we feel that it has merit but does not fully meet PLOS Neglected Tropical Diseases's publication criteria as it currently stands. Therefore, we invite you to submit a revised version of the manuscript that addresses the points raised during the review process. Please submit your revised manuscript within 30 days May 17 2025 11:59PM. If you will need more time than this to complete your revisions, please reply to this message or contact the journal office at plosntds@plos.org. Please include the following items when submitting your revised manuscript: * A rebuttal letter that responds to each point raised by the editor and reviewer(s). You should upload this letter as a separate file labeled 'Response to Reviewers '. This file does not need to include responses to any formatting updates and technical items listed in the 'Journal Requirements' section below.* A marked-up copy of your manuscript that highlights changes made to the original version. You should upload this as a separate file labeled 'Revised Manuscript with Track Changes '.* An unmarked version of your revised paper without tracked changes. You should upload this as a separate file labeled 'Manuscript '. If you would like to make changes to your financial disclosure, competing interests statement, or data availability statement, please make these updates within the submission form at the time of resubmission. Guidelines for resubmitting your figure files are available below the reviewer comments at the end of this letter. We look forward to receiving your revised manuscript. Kind regards, Sutas Suttiprapa, Ph.D.Academic EditorPLOS Neglected Tropical Diseases Krystyna CwiklinskiSection EditorPLOS Neglected Tropical Diseases

Shaden Kamhawi

co-Editor-in-Chief

Paul Brindley

co-Editor-in-Chief

**Additional Editor Comments :** The authors have addressed all the points raised by the three reviewers. However, there are still some typographical errors and language issues that need to be addressed before final acceptance. **Journal Requirements:**

At this stage, the following Authors/Authors require contributions: Yoonho Cho, Jungim Lee, Humphrey Mazigo, Leah Elisha Salamba, and Seungman Cha. Please ensure that the full contributions of each author are acknowledged in the "Add/Edit/Remove Authors" section of our submission form.

2) The file inventory includes files for Figures 1a, and 1b. We would recommend either combining these into a single Figure 1.tiff file with separate internal panels, or renumbering them as individual figures, as we are not able to publish multiple components of a single figure as separate files.

**Reviewers' comments:** Reviewer's Responses to Questions

**Key Review Criteria Required for Acceptance?**

**Methods**

-Are the objectives of the study clearly articulated with a clear testable hypothesis stated?

-Is the study design appropriate to address the stated objectives?

-Is the population clearly described and appropriate for the hypothesis being tested?

-Is the sample size sufficient to ensure adequate power to address the hypothesis being tested?

-Were correct statistical analysis used to support conclusions?

-Are there concerns about ethical or regulatory requirements being met?

Reviewer #1: The concerns I raised regarding key parts of the methods have been addressed.

Reviewer #2: (No Response)

Reviewer #3: (No Response)

**Results**

-Does the analysis presented match the analysis plan?

-Are the results clearly and completely presented?

-Are the figures (Tables, Images) of sufficient quality for clarity?

Reviewer #1: The results section reads well and is now in good shape for publication.

Reviewer #2: (No Response)

Reviewer #3: Please correct any typing errors in the text and figures, such as the legend in Fig. 2. The red circle should be 40% or above.

**Conclusions**

-Are the conclusions supported by the data presented?

-Are the limitations of analysis clearly described?

-Do the authors discuss how these data can be helpful to advance our understanding of the topic under study?

-Is public health relevance addressed?

Reviewer #1: The conclusion are well-grounded on the results and the discussion.

Reviewer #2: (No Response)

Reviewer #3: (No Response)

**Editorial and Data Presentation Modifications?**

Reviewer #1: (No Response)

Reviewer #2: The authors have addressed all my queries. However, I think it will be helpful for the manuscript to undergo proof-reading for language clarity prior to publication.

Reviewer #3: (No Response)

**Summary and General Comments**

Reviewer #1: The entire manuscript is now publishable. Hence, I recommend its acceptance.

Reviewer #2: (No Response)

Reviewer #3: (No Response)

PLOS authors have the option to publish the peer review history of their article (what does this mean? ). If published, this will include your full peer review and any attached files.

**Do you want your identity to be public for this peer review?** For information about this choice, including consent withdrawal, please see our Privacy Policy .

Reviewer #1: No

Reviewer #2: No

Reviewer #3: No

**Figure resubmission:** While revising your submission, please upload your figure files to the Preflight Analysis and Conversion Engine (PACE) digital diagnostic tool, https://pacev2.apexcovantage.com/. PACE helps ensure that figures meet PLOS requirements. To use PACE, you must first register as a user. Registration is free. Then, login and navigate to the UPLOAD tab, where you will find detailed instructions on how to use the tool. If you encounter any issues or have any questions when using PACE, please email PLOS at figures@plos.org. Please note that Supporting Information files do not need this step. If there are other versions of figure files still present in your submission file inventory at resubmission, please replace them with the PACE-processed versions.**Reproducibility:** To enhance the reproducibility of your results, we recommend that authors of applicable studies deposit laboratory protocols in protocols.io, where a protocol can be assigned its own identifier (DOI) such that it can be cited independently in the future. Additionally, PLOS ONE offers an option to publish peer-reviewed clinical study protocols. Read more information on sharing protocols at https://plos.org/protocols?utm_medium=editorial-email&utm_source=authorletters&utm_campaign=protocols

---

## [Decision Letter · Decision Letter 2]

Dear Professor Cha,

We are pleased to inform you that your manuscript 'Effects of an integrated intervention on schistosomiasis prevalence in a rural area of Tanzania' has been provisionally accepted for publication in PLOS Neglected Tropical Diseases.

Best regards,

Sutas Suttiprapa, Ph.D.

Academic Editor

Krystyna Cwiklinski

Section Editor

Shaden Kamhawi

co-Editor-in-Chief

Paul Brindley

co-Editor-in-Chief

Reviewer's Responses to Questions

**Key Review Criteria Required for Acceptance?**

**Methods**

-Are the objectives of the study clearly articulated with a clear testable hypothesis stated?

-Is the study design appropriate to address the stated objectives?

-Is the population clearly described and appropriate for the hypothesis being tested?

-Is the sample size sufficient to ensure adequate power to address the hypothesis being tested?

-Were correct statistical analysis used to support conclusions?

-Are there concerns about ethical or regulatory requirements being met?

Reviewer #2: (No Response)

**Results**

-Does the analysis presented match the analysis plan?

-Are the results clearly and completely presented?

-Are the figures (Tables, Images) of sufficient quality for clarity?

Reviewer #2: (No Response)

**Conclusions**

-Are the conclusions supported by the data presented?

-Are the limitations of analysis clearly described?

-Do the authors discuss how these data can be helpful to advance our understanding of the topic under study?

-Is public health relevance addressed?

Reviewer #2: (No Response)

**Editorial and Data Presentation Modifications?**

Reviewer #2: (No Response)

**Summary and General Comments**

Reviewer #2: (No Response)

PLOS authors have the option to publish the peer review history of their article (what does this mean? ). If published, this will include your full peer review and any attached files.

**Do you want your identity to be public for this peer review?** For information about this choice, including consent withdrawal, please see our Privacy Policy .

Reviewer #2: No

---

## [Editor Report · Acceptance letter]

Dear Professor Cha,

We are delighted to inform you that your manuscript, "Effects of an integrated intervention on schistosomiasis prevalence in a rural area of Tanzania," has been formally accepted for publication in PLOS Neglected Tropical Diseases.

Best regards,

Shaden Kamhawi

co-Editor-in-Chief

Paul Brindley

co-Editor-in-Chief
